# Correlation Study Between Neoadjuvant Chemotherapy Response and Long-Term Prognosis in Breast Cancer Based on Deep Learning Models

**DOI:** 10.3390/diagnostics15212763

**Published:** 2025-10-31

**Authors:** Ke Wang, Yikai Luo, Peng Zhang, Bing Yang, Yubo Tao

**Affiliations:** 1Breast Surgery Department, Second Affiliated Hospital, Zhejiang University School of Medicine, Hangzhou 310009, China; 2State Key Lab of CAD&CG, College of Computer Science and Technology, Zhejiang University, Hangzhou 310058, China

**Keywords:** breast cancer, neoadjuvant chemotherapy, artificial intelligence, recurrence and metastasis, prognosis prediction

## Abstract

**Background****:** The pathological response to neoadjuvant chemotherapy (NAC) is an established predictor of long-term outcomes in breast cancer. However, conventional binary assessment based solely on pathological complete response (pCR) fails to capture prognostic heterogeneity across molecular subtypes. This study aimed to develop an interpretable deep learning model that integrates multiple clinical and pathological variables to predict both recurrence and metastasis development following NAC treatment. **Methods:** We conducted a retrospective analysis of 832 breast cancer patients who received NAC between 2013 and 2022. The analysis incorporated five key variables: tumor size changes, nodal status, Ki-67 index, Miller–Payne grade, and molecular subtype. A Multi-Layer Perceptron (MLP) model was implemented on the PyTorch platform and systematically benchmarked against SVM, Random Forest, and XGBoost models using five-fold cross-validation. Model performance was assessed by calculating the area under the curve (AUC), accuracy, precision, recall, and F1-score, and by analyzing confusion matrices. **Results:** The MLP model achieved AUC values of 0.86 (95% CI: 0.82–0.93) for HER2-positive cases, 0.82 (95% CI: 0.70–0.92) for triple-negative cases, and 0.76 (95% CI: 0.66–0.82) for HR+/HER2-negative cases. SHAP analysis identified post-NAC tumor size, Ki-67 index, and Miller–Payne grade as the most influential predictors. Notably, patients who achieved pCR still had a 12% risk of developing recurrence, highlighting the necessity for ongoing risk assessment beyond binary response evaluation. **Conclusions:** The proposed deep learning system provides precise and interpretable risk assessment for NAC patients, facilitating individualized treatment approaches and post-treatment monitoring plans.

## 1. Background

Breast cancer is the most common malignant tumor among women worldwide, with both incidence and mortality continuing to rise. Neoadjuvant chemotherapy (NAC), which is an important component of the standard treatment regimen for locally advanced breast cancer, plays a key role in tumor downstaging, improving breast conservation rates, achieving early control of micrometastases, and evaluating chemotherapy sensitivity [1,2,3]. However, clinical outcomes after NAC remain highly heterogeneous, ranging from pathological complete response (pCR) to disease progression, which directly affects long-term prognosis [4,5].

Currently, prognostic evaluation primarily relies on TNM staging, molecular subtype classification, and clinicopathological indicators. Meta-analyses have shown that patients who achieve pathological complete response (pCR) have different survival rates depending on their cancer subtype [6,7]. Cortazar et al. reported that triple-negative breast cancer (TNBC) patients achieving pCR still face poorer long-term survival compared to hormone receptor-positive patients [8]. These observations indicate that relying on single-source assessment methods fails to capture the full diversity of breast cancer, necessitating multi-source evaluation frameworks for comprehensive assessment.

To bridge this gap, recent research has increasingly applied artificial intelligence (AI) and deep learning (DL) in oncology to uncover complex nonlinear relationships across multimodal datasets [9,10,11]. In the field of breast cancer research, several recent studies have investigated AI systems for predicting treatment responses [12,13]. Kim et al. and He et al. applied classical machine learning models using clinicopathological data for pCR prediction, demonstrating feasibility while revealing the performance limitations of simpler models [14,15]. Similarly, research by Mao et al. implemented multi-modal feature optimization techniques [16], while Zhou et al. combined ultrasound radiomics with clinical data in a multicenter trial to enhance predictive performance [17].

However, these existing approaches present important limitations that our study seeks to address. Most are constrained by their reliance on single time-point measurements, radiomics-only pipelines, or the absence of systematic benchmarking across diverse ML methods. Furthermore, predicting long-term therapeutic outcomes such as recurrence or metastasis is substantially more complex than predicting pCR, as it requires modeling dynamic interactions between baseline characteristics, treatment-induced changes, and longitudinal outcomes.

Our study addresses these gaps by proposing a comprehensive deep learning framework integrating pre- and post-treatment variables to predict long-term recurrence risk after NAC. Specifically, our study is designed to address three key scientific questions: (1) How can we quantitatively model the nonlinear relationship between tumor size dynamics and survival outcomes? (2) How can we identify patient subgroups who achieve pCR but remain at elevated risk of recurrence? and (3) How can we leverage the prognostic value of Miller–Payne grading beyond binary response classification?

The novelty of our work lies in three major aspects. First, unlike previous studies, our method uses pre-treatment and post-treatment data to directly forecast long-term recurrence and metastasis risk rather than focusing solely on short-term pCR prediction. Second, the model identifies patients who achieve pCR but face high recurrence risk, thereby extending beyond standard clinical assessment methods. Third, our research rigorously evaluates the performance of our Multi-layer Perceptron (MLP) model through comprehensive benchmarking against SVM, Random Forest, and XGBoost to demonstrate deep learning’s superior value in this predictive task.

## 2. Materials and Methods

### 2.1. Study Population and Design

A total of 904 breast cancer patients who received neoadjuvant chemotherapy (NAC) and subsequent surgery between 2013 and 2022 were initially screened, of whom 832 were included in the final analysis after applying the inclusion and exclusion criteria. Inclusion criteria were: (1) age ≥18 years; (2) invasive breast cancer confirmed by core needle biopsy; (3) completion of a standard NAC regimen; and (4) availability of complete clinicopathological and follow-up data. Exclusion criteria were: (1) distant metastasis at initial diagnosis; (2) prior history of malignancy; (3) discontinuation of the planned NAC regimen; (4) pregnancy-associated breast cancer; and (5) male breast cancer. To ensure methodological transparency, all clinical records were independently reviewed by two oncologic surgeons, and ambiguous data points were verified by chart review. This study was approved by the Ethics Committee of the Second Affiliated Hospital of Zhejiang University School of Medicine (IRB-2021-930), and the requirement for informed consent was waived due to its retrospective nature.

### 2.2. Treatment Protocol

All patients received standard NAC regimens based on taxanes in combination with anthracyclines or platinum agents. Patients with HER2-positive cancer received trastuzumab, with or without pertuzumab. Those with hormone receptor-positive disease initiated endocrine therapy after completing chemotherapy. Surgical procedures post-NAC included either breast-conserving surgery or mastectomy with sentinel lymph node biopsy (SLNB) and/or axillary lymph node dissection (ALND), depending on clinical staging and response. All treatment strategies were determined according to the latest NCCN Breast Cancer Guidelines and institutional consensus protocols to ensure treatment consistency.

### 2.3. Dataset Description

The dataset comprised patient demographics, tumor characteristics, treatment indicators, and long-term outcomes. Key features included age, baseline tumor size, lymph node status, hormone receptor (HR) and HER2 expression, Ki-67 index, Miller–Payne grade, surgical type, and post-NAC residual tumor characteristics. Recurrence status within five years was defined as the binary outcome for supervised classification. The output label was binary (recurrence/metastasis within five years = 1; disease-free = 0), confirming that the problem formulation was a supervised binary classification task. Categorical variables were appropriately encoded, and continuous variables were standardized using z-scores prior to modeling. Missing data (<5%) were imputed using multiple imputation by chained equations (MICE).

### 2.4. Clinicopathological Assessment

Tumor specimens were obtained via ultrasound-guided core needle biopsy. Molecular markers were assessed by immunohistochemistry (IHC). The definition of hormone receptor (HR) positivity required ≥1% nuclear staining for estrogen receptor (ER) or progesterone receptor (PR). The determination of HER2 status followed ASCO/CAP guidelines which classified patients into three groups based on IHC results: 0/1+ as negative and 3+ as positive and 2+ required FISH confirmation. The Ki-67 proliferation index values were divided into two groups based on 20% to distinguish between high and low expression levels. The patients were divided into three molecular subtypes which included HR+/HER2-negative and HER2+ and triple-negative. To minimize interobserver bias, all pathological slides were independently reviewed by two board-certified pathologists blinded to patient outcomes, with disagreements resolved through consensus.

### 2.5. Response Evaluation and Follow-Up

The assessment of treatment response included both radiological measurements and pathological evaluations. The RECIST 1.1 criteria served for radiological assessment to measure tumor maximum diameter changes during NAC treatment. The Miller–Payne grading system (Grade 1–5) served as the method for pathological evaluation of treatment response. The definition of Pathological Complete Response (pCR) required no remaining invasive cancer in breast tissue and axillary lymph nodes (ypT0/is ypN0). Miller–Payne grade and continuous tumor reduction ratio were used as predictive measures because these metrics offer more detailed information than the binary pCR/non-pCR classification system. The primary endpoint was disease-free survival (DFS), defined as the interval from surgery date to first recurrence/metastasis or last follow-up. Overall survival (OS) was also recorded as a secondary endpoint.

### 2.6. Feature Selection and Model Development

Feature importance was evaluated using a combination of Random Forest-based selection and clinical expert consensus. The final feature set included molecular subtype, histological grade, tumor size reduction ratio, lymph node status before and after NAC, Miller–Payne grade, Ki-67 index, and surgical type. These features were selected based on their statistical importance and established clinical relevance to breast cancer prognosis.

The deep learning model was developed using Python 3.8 and PyTorch 1.9. The Multi-layer Perceptron (MLP) architecture comprised an input layer with dimensionality matching the number of selected features, three fully connected hidden layers with 16, 32, and 16 neurons, respectively, ReLU activation functions, and a dropout rate of 0.2 for regularization, followed by an output layer with 2 nodes using Softmax activation for binary classification.

To enhance model generalizability, five-fold cross-validation was performed. Each fold involved re-initialization of model weights and independent training-validation splits. Early stopping was applied with a patience of 15 epochs, and learning curves were visually inspected to confirm convergence.

The model was trained with the Adam optimizer at a learning rate of 0.005, a batch size of 32, and a cross-entropy loss function. Hyperparameters were optimized via grid search (learning rate: 0.001–0.01; dropout: 0.1–0.3; hidden units: 16–32–16). For comparison, three benchmark models were implemented: Support Vector Machine (SVM) with a radial basis function kernel, Random Forest with 100 estimators, and XGBoost with default parameters for binary classification.

### 2.7. Statistical Analysis

Model performance was evaluated using the area under the receiver operating characteristic curve (AUC) with 95% confidence intervals [18], along with accuracy, precision, recall, sensitivity, specificity, positive predictive value (PPV), negative predictive value (NPV), and F1 score. Additionally, probabilistic predictions were assessed using mean absolute error (MAE) and mean squared error (MSE). The optimal risk probability thresholds for prediction were determined using the Youden index. The DeLong test was used to compare AUCs across models. All statistical tests were two-sided, and a *p* value < 0.05 was considered statistically significant.

## 3. Results

### 3.1. Patient Characteristics

A total of 904 breast cancer patients who received NAC were initially screened for this study. After excluding 72 cases with incomplete clinical data or loss to follow-up, a total of 832 patients were included in the final analysis. Among these patients, 208 (25.0%) achieved pathological complete response (pCR), whereas 624 (75.0%) had residual invasive disease and/or axillary lymph node metastases. All patients received standardized NAC regimens based on anthracyclines and/or taxanes, and HER2-positive cases were treated with trastuzumab with or without pertuzumab. With respect to surgery, 193 patients (23.2%) underwent breast-conserving surgery and 639 patients (76.8%) underwent modified radical mastectomy. The median follow-up time was 60.2 months, during which 135 patients (16.2%) developed recurrence or metastasis, yielding a 5-year DFS rate of 83.8% and an overall survival rate of 87.6%.

The baseline clinicopathological characteristics of the patients, stratified by pCR status, are summarized in Table 1. Statistical comparisons revealed significant differences between the pCR and non-pCR groups in several key variables, including molecular subtype, clinical stage, Ki-67 expression, post-NAC lymph node status, type of surgery, and Miller–Payne grade (all *p* < 0.05). In contrast, age was not significantly different between the two groups (median 51 years in non-pCR vs. 52 years in pCR, *p* > 0.05). As anticipated, patients with HER2-positive and triple-negative breast cancer (TNBC) subtypes exhibited significantly higher pCR rates, whereas HR+/HER2− patients were less responsive to NAC. Furthermore, high Ki-67 expression (≥20%) was more prevalent in the pCR group (90.4%) compared to the non-pCR group (79.8%).

### 3.2. Model Performance Evaluation

The predictive features for the model were finalized through a combination of Random Forest-based importance ranking and consensus from clinical experts. The final set included molecular subtype, histological grade, tumor size reduction ratio, lymph node status before and after NAC, Miller–Payne grade, Ki-67 index, and surgical type.

To predict recurrence and metastasis risk, we developed a Multi-layer Perceptron (MLP) model to integrate these features for prognosis prediction. The architecture of this model is schematically depicted in Figure 1. It was designed with input neurons, corresponding to the selected clinical features, followed by three fully connected hidden layers (16, 32, and 16 neurons, respectively) with ReLU activation and Dropout regularization, culminating in a 2-neuron output layer with Softmax activation for binary classification.

Under five-fold cross-validation, the proposed MLP model demonstrated robust and stable performance. It achieved an AUC of 0.86 (95% CI: 0.81–0.92), an accuracy of 0.86, a sensitivity (recall) of 0.85, and an F1-score of 0.84,indicating strong discriminative ability for predicting recurrence and metastasis following NAC (Figure 2). SHAP analysis further confirmed that post-NAC residual tumor size, Ki-67 index, and Miller–Payne grade were the most influential predictors, underscoring their central role in prognostic stratification.

Benchmarking results (Table 2) showed that the deep learning-based MLP model consistently outperformed traditional machine learning algorithms, including SVM, Random Forest, and XGBoost. Specifically, the MLP achieved the highest AUC (0.86, 95% CI: 0.81–0.92), accuracy (0.86), and F1-score (0.84), demonstrating superior discriminative ability in predicting recurrence and metastasis risk after NAC. In comparison, the best-performing traditional ML model (XGBoost) yielded an AUC of 0.81 and F1-score of 0.78, while SVM and RF models performed slightly lower (AUC range: 0.76–0.80). These findings underscore the advantage of deep learning in capturing nonlinear relationships across heterogeneous clinical and pathological variables.

### 3.3. Subgroup Analysis

#### 3.3.1. Molecular Subtypes

The model exhibited strong predictive capability across subtypes, though with variations. The HER2-positive group demonstrated the best performance (AUC 0.86, 95% CI: 0.82–0.93), followed by the triple-negative group (AUC 0.82, 95% CI: 0.70–0.92) and the HR+/HER2-negative group (AUC 0.76, 95% CI: 0.66–0.82) (Figure 3). This pattern mirrors the clinical observation that HER2+ and TNBC subtypes show more distinct treatment responses and thus allow better model discrimination.

#### 3.3.2. Surgical Approaches

When stratified by surgery, breast-conserving patients achieved slightly higher predictive accuracy (AUC 0.88, 95% CI: 0.79–0.95) compared with mastectomy patients (AUC 0.85, 95% CI: 0.70–0.89) (Figure 4). In the breast-conserving group, the model reached Sensitivity 0.88 (95% CI: 0.80–0.96) and Specificity 0.86 (95% CI: 0.78–0.94), indicating superior discrimination. In contrast, mastectomy patients demonstrated balanced Sensitivity (0.78, 95% CI: 0.72–0.83) and Specificity (0.75, 95% CI: 0.69–0.81). This may reflect differences in tumor burden and pathological assessment depth between surgical subgroups.

#### 3.3.3. Pathological Response Groups

Prediction accuracy was broadly comparable between non-pCR (AUC 0.75, 95% CI: 0.62–0.88) and pCR patients (AUC 0.79, 95% CI: 0.66–0.82) (Figure 5). Notably, both groups demonstrated high sensitivity, with the pCR group at 0.84 (95% CI: 0.73–0.95) and the non-pCR group at 0.89 (95% CI: 0.83–0.94), though specificity was relatively lower (pCR group 0.63, 95% CI: 0.49–0.77; non-pCR group 0.62, 95% CI: 0.53–0.71). This indicates that while the model reliably detects high-risk patients, it may overestimate recurrence risk in some lower-risk cases.

### 3.4. Predictive Factors and High-Risk Identification

A critical and clinically relevant finding of this study was that 25 out of the 208 patients (12.0%) who achieved a pCR subsequently developed recurrence or metastasis during the follow-up period. This observation underscores the significant limitation of relying solely on binary pCR status for prognostic stratification and highlights the necessity for more refined risk assessment tools. To elucidate the factors contributing to recurrence in this ostensibly low-risk pCR subgroup, we performed a SHAP analysis on these patients specifically (Figure 6). The analysis quantified the contribution of various factors, identifying the Ki-67 proliferation index as the most influential predictor (mean SHAP value: 0.129), followed by molecular subtype (0.064), surgical approach (0.025), pre-NAC tumor size, age, and clinical stage. These results powerfully demonstrate the utility of integrating post-treatment biological markers (like Ki-67) and comprehensive clinical profiles to identify a residual high-risk population even among those who achieve a pCR, thereby enabling more tailored and vigilant post-treatment management.

Overall, the findings from the results section demonstrate that the proposed MLP framework not only matches but significantly surpasses conventional machine learning models in predictive accuracy for long-term recurrence risk after NAC. Furthermore, its interpretability through SHAP analysis provides clinically actionable insights, facilitating individualized follow-up strategies and adjuvant therapy planning.

## 4. Discussion

This study demonstrates that integrating multi-modal clinicopathological data through a deep learning framework significantly enhances the prediction accuracy of both neoadjuvant chemotherapy (NAC) response and long-term prognosis in breast cancer. The combination of baseline clinical data with molecular subtypes and histopathological markers and post-NAC pathological grading enables our approach to detect complex time-dependent relationships which standard models fail to recognize. The Multi-layer Perceptron model achieved robust predictive performance (AUC 0.86, Accuracy 0.86, and F1-score 0.84), supporting its potential for clinical translation.

Our results represent a meaningful advancement over prior research. Zhou et al. achieved AUC values between 0.75 and 0.80 using ultrasound radiomics and classical machine learning methods [17], while Mao et al. obtained similar results with their multimodal approach [16]. Notably, a recent deep learning system developed by Li et al. also demonstrated high efficacy in predicting pCR from dynamic contrast-enhanced magnetic resonance images, further validating the role of AI in this domain [19]. Compared to these studies, our work uniquely integrates pre- and post-treatment clinical parameters (Miller–Payne grade and Ki-67 index) within a unified temporal framework, thereby addressing the research gap between short-term response prediction and long-term recurrence forecasting. The performance metrics were also computed systematically using a confusion-matrix framework, with explicit formulas added in the Methods Section. This inclusion not only clarifies the derivation of accuracy, precision, recall, and F1 scores but also enhances transparency for reproducibility. Each model’s confusion matrix further visualizes prediction patterns and error distributions, aiding intuitive interpretation for clinicians.

From a mechanistic and clinical perspective, our findings offer valuable insights. SHAP analysis identified post-NAC tumor size, Ki-67 proliferation index, and Miller–Payne grade as the strongest predictors of recurrence and metastasis. These features reflect residual tumor burden and proliferative activity—two key biological dimensions central to post-treatment risk assessment. Importantly, our study reinforces that continuous measures of tumor shrinkage provide more prognostic information than binary pCR classification alone, which is consistent with previous findings by Symmans et al. [20,21] and von Minckwitz et al. [22]. This supports the development of advanced risk assessment models that use detailed response measurements to create personalized treatment plans.

Clinically, this work offers several translational advantages. First, the model’s modular design facilitates the incorporation of additional data types—such as radiomic, genomic, or proteomic features—enabling dynamic risk updates throughout treatment. Second, its interpretability via SHAP visualization bridges the gap between AI prediction and clinical reasoning, enhancing its acceptability for physician-assisted decision-making. Moreover, the model demonstrates stable performance across different molecular subtypes, especially in HER2-positive cases, indicating its potential to support treatment decisions for diverse patient groups.

Future research should focus on (1) prospective validation in multicenter cohorts; (2) integration of multi-omics features to enhance predictive power; (3) implementation within clinical decision support systems to guide therapy; and (4) development of automated pipelines to minimize manual preprocessing and advance toward real-time, end-to-end clinical prediction.

In conclusion, this work fills a critical research gap by providing a transparent, comprehensively benchmarked deep-learning framework for predicting long-term prognosis after NAC. Beyond conventional pCR classification, this approach identifies residual high-risk patients and provides a foundation for personalized treatment and follow-up strategies in precision breast cancer management.

## Figures and Tables

**Figure 1 diagnostics-15-02763-f001:**
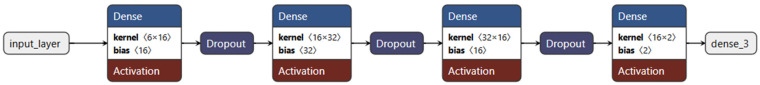
Schematic diagram of the proposed Multi-Layer Perceptron (MLP) model architecture. The model takes clinical features as input. The data flows through three fully connected (Dense) hidden layers with 16, 32, and 16 neurons, respectively, each followed by a ReLU activation function and a Dropout layer for regularization. The final output layer consists of 2 neurons with Softmax activation, producing a probability for ‘Recurrence/Metastasis’ or ‘Disease-Free’. The kernel weight matrix and bias vector dimensions for each layer are explicitly noted.

**Figure 2 diagnostics-15-02763-f002:**
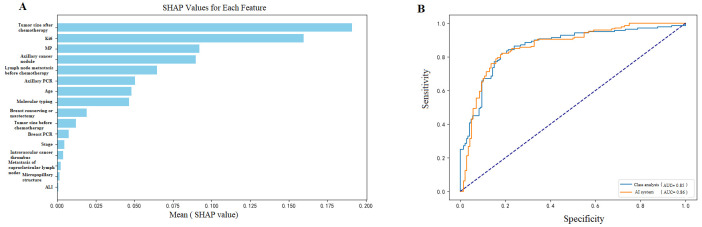
Model performance visualization showing: (**A**) SHAP value analysis of predictive features, where post-chemotherapy tumor size, Ki-67, and Miller–Payne grade emerged as the top contributing factors (mean SHAP values: 0.16, 0.17, and 0.09, respectively), and (**B**) ROC demonstrating robust model discrimination with AUC of 0.86 (95% CI: 0.81–0.92; sensitivity: 0.85, 95% CI: 0.81–0.89). AUC = area under the receiver operating curve.

**Figure 3 diagnostics-15-02763-f003:**
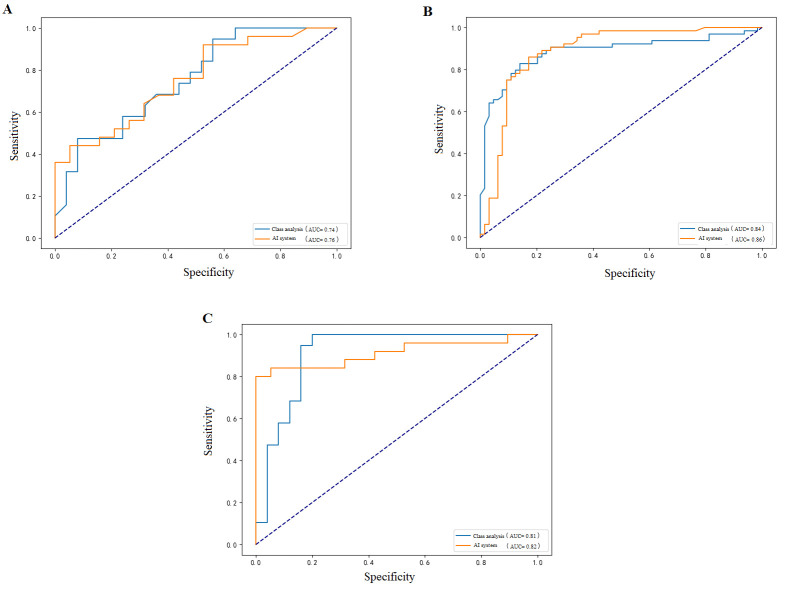
Receiver operating characteristic (ROC) curves showing the model’s predictive performance across different molecular subtypes: (**A**) HR+/HER2- subtype with AUC of 0.76 (95% CI: 0.66–0.82), (**B**) HER2-positive subtype demonstrating the best performance with AUC of 0.86 (95% CI: 0.82–0.93), and (**C**) triple-negative subtype with AUC of 0.82 (95% CI: 0.70–0.93), reflecting the distinct predictive capabilities of the model across different breast cancer molecular subtypes.

**Figure 4 diagnostics-15-02763-f004:**
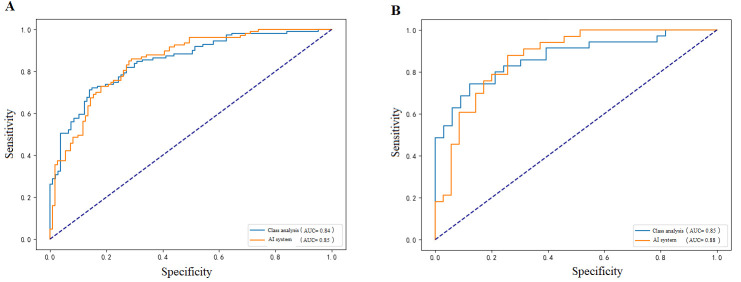
ROC curves illustrating the model’s predictive performance in different surgical groups: (**A**) mastectomy group with AUC of 0.85 (95% CI: 0.70–0.89), and (**B**) breast-conserving surgery group with slightly better performance (AUC of 0.88, 95%CI: 0.79–0.95).

**Figure 5 diagnostics-15-02763-f005:**
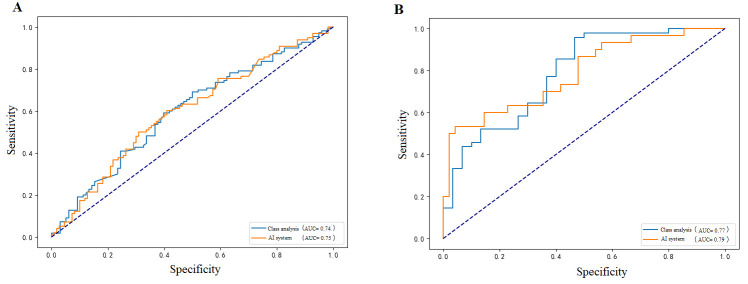
ROC curves comparing model performance between different pathological response groups: (**A**) non-pCR group showing moderate predictive ability with AUC of 0.75 (95% CI: 0.62–0.88), while (**B**) pCR group demonstrating better discrimination with AUC of 0.79 (95% CI: 0.66–0.82), indicating the model’s enhanced capability in risk stratification among patients achieving pathological complete response.

**Figure 6 diagnostics-15-02763-f006:**
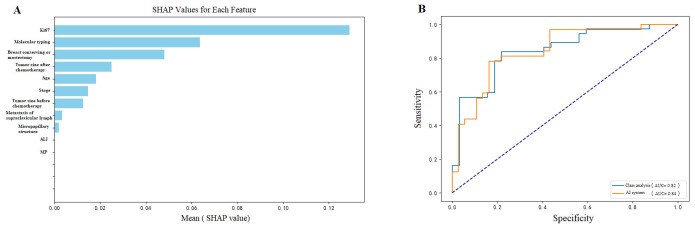
Model performance visualization showing: (**A**) SHAP value analysis revealing Ki-67 as the strongest predictor (SHAP value: 0.129), followed by molecular typing (0.064) and surgical approach (0.025), and (**B**) ROC curves demonstrates robust model discrimination with AUC of 0.84 (95% CI: 0.81–0.89), indicating strong predictive capability for patient outcomes. AUC = area under the receiver operating curve.

**Table 1 diagnostics-15-02763-t001:** Baseline Clinicopathological Characteristics [*n* (%)].

Characteristics	Non-pCR	pCR	*p*-Value
**Age** **(year)**			NS
Median	51	52	
**Cancer subtype**			*p* < 0.05
HR+/HER2-	307	20	
HER2+	106	32	
Triple-Negative	211	156	
**Clinical T stage, n (%)**			NS
cT1–2	5	3	
cT3–4	616	208	
**Clinical stage**			*p* < 0.05
Stage I	10	2	
Stage II	369	161	
Stage III	245	45	
**Ki-67 status (≥20%)**			*p* < 0.05
Positive	497	188	
Negative	126	21	
**Post-NAC lymph node status (ypN)**			<0.05
Positive	412 (66.0%)	0 (0.0%)	
Negative	212 (34.0%)	208 (100%)	
**Type of surgery**			<0.05
Mastectomy	512 (82.1%)	127 (61.1%)	
Breast-conserving surgery	112 (17.9%)	81 (38.9%)	
**Miller–Payne grade**			<0.05
0–4	624 (100%)	0 (0.0%)	
5	0 (0.0%)	208 (100%)	
Long-term Outcomes Total (N = 832)			
5-Year DFS Rate	697 (83.8%)		
Recurrence/Metastasis	135 (16.2%)		
5-Year OS Rate	725 (87.6%)		

Abbreviations: pCR, Pathological Complete Response; TN, Triple-Negative; DFS, Disease-Free Survival; OS, Overall Survival; NS, Not Significant.

**Table 2 diagnostics-15-02763-t002:** Comparison of predictive performance across different models.

Model	AUC (95% CI)	Accuracy	Precision	Recall	F1-Score	MAE	MSE
MLP	0.86 (0.81–0.92)	0.86	0.85	0.85	0.84	0.124	0.056
XGBoost	0.81 (0.77–0.86)	0.81	0.79	0.78	0.78	0.141	0.068
Random Forest	0.79 (0.74–0.84)	0.78	0.77	0.75	0.76	0.153	0.075
SVM	0.76 (0.70–0.82)	0.75	0.73	0.74	0.74	0.162	0.081

## Data Availability

All data generated or analyzed during this study are included in this published article.

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
